# Nanothin Conformal Coating with Poly(N-vinylpyrrolidone) and Tannic Acid (PVPON/TA) Preserves Murine and Human Pancreatic Islets Function

**DOI:** 10.3390/pharmaceutics15041137

**Published:** 2023-04-04

**Authors:** Kateryna Polishevska, Sandra Kelly, Purushothaman Kuppan, Karen L. Seeberger, Saloni Aggarwal, Joy Paramor, Larry D. Unsworth, Hubert M. Tse, Gregory S. Korbutt, Andrew R. Pepper

**Affiliations:** 1Alberta Diabetes Institute, University of Alberta, Edmonton, AB T6G 2T9, Canada; 2Department of Surgery, Faculty of Medicine and Dentistry, University of Alberta, Edmonton, AB T6G 2R3, Canada; 3Department of Chemical and Materials Engineering, University of Alberta, Edmonton, AB T6G 2E1, Canada; 4Department of Microbiology, Molecular Genetics, and Immunology, University of Kansas Medical Center, Kansas City, KS 66160, USA

**Keywords:** islet transplantation, type 1 diabetes, conformal coating, encapsulation, allotransplantation, human islets

## Abstract

Beta cell replacement therapies can restore glycemic control to select individuals living with type 1 diabetes. However, the obligation of lifelong immunosuppression restricts cell therapies from replacing exogenous insulin administration. Encapsulation strategies can reduce the inherent adaptive immune response; however, few are successfully translated into clinical testing. Herein, we evaluated if the conformal coating of islets with poly(N-vinylpyrrolidone) (PVPON) and tannic acid (TA) (PVPON/TA) could preserve murine and human islet function while conferring islet allograft protection. In vitro function was evaluated using static glucose-stimulated insulin secretion, oxygen consumption rates, and islet membrane integrity. In vivo function was evaluated by transplanting human islets into diabetic immunodeficient B6.129S7-Rag^1tm1Mom^/J (Rag^-/-^) mice. The immunoprotective capacity of the PVPON/TA-coating was assessed by transplanting BALB/c islets into diabetic C57BL/6 mice. Graft function was evaluated by non-fasting blood glucose measurements and glucose tolerance testing. Both coated and non-coated murine and human islets exhibited indistinguishable in vitro potency. PVPON/TA-coated and control human islets were able to restore euglycemia post-transplant. The PVPON/TA-coating as monotherapy and adjuvant to systemic immunosuppression reduced intragraft inflammation and delayed murine allograft rejection. This study demonstrates that PVPON/TA-coated islets may be clinically relevant as they retain their in vitro and in vivo function while modulating post-transplant immune responses.

## 1. Introduction

Diabetes is a major health issue that has reached alarming levels: worldwide, nearly half a billion people are living with diabetes [1]. Disrupted production of insulin by the pancreatic islets of Langerhans is considered the central cause of all forms of diabetes [2]. Type 1 diabetes (T1D; ~15% of cases) is caused by an autoimmune reaction toward insulin-producing beta cells of the pancreas. The most common treatment of T1D is subcutaneous injections of insulin to manage blood glucose levels; but, glycemic levels can fluctuate considerably outside the physiological range. Conversely, recurrent hypoglycemia is no less threatening since it can advance to life-threatening hypoglycemia unawareness [3]. Recurrent hypoglycemia and hypoglycemia unawareness are often neglected complications of diabetes since most clinical guidelines are directed toward managing hyperglycemia. Nevertheless, life-threatening hypoglycemia unawareness remains common (up to 40%) [3,4], which negatively impacts patients’ quality of life (e.g., psychosocial behavior), and increases disease-related morbidity (e.g., cardiovascular disease) and mortality (e.g., up 10% succumb to severe hypoglycemia) [5,6]. The limitations of using exogenous insulin to maintain physiological glucose levels have led to the development of alternative therapies, including the insulin pump, whole organ transplantation, and islet transplantation.

Pancreatic islet transplantation effectively eliminates severe hypoglycemic events and restores glycemic control in patients with T1D. Despite considerable improvements in clinical islet transplantation [7], the transplant approaches are mainly directed at patients suffering from severe hypoglycemia and hypoglycemia unawareness [8]. Recent data have demonstrated that median islet graft survival is approximately 6 years while insulin independence declines from approximately 60% at 1 year to 20% at 5 years and less than 10% at 20 years post-transplantation [9]. While both auto- and allo-immune mediated rejection potentiates long-term islet graft failure, accumulative evidence indicates that acute islet cell death post-transplantation severely compromises engraftment [10]. As a repercussion, multiple pancreas donors and lifelong systemic immunosuppression are necessary to achieve insulin independence; restricting beta cell replacement therapy to a narrow subset of individuals living with T1D.

The necessity for immunosuppression may be reduced through islet encapsulation. Microencapsulation involves coating one or several islets within a thin spherical layer of protective materials where the surface-to-area/volume ratio can be altered to maximize the efficient oxygen and nutrient exchange required for islet survival and efficacy [11]. A disadvantage of conventional microencapsulation utilizing alginate, or another biocompatible material, is that the resulting capsules vary between 350 to 500 μm in diameter. Consequently, this increases the transplant volume, restricts transplant sites and anatomical locations (e.g., portal vein), increases the distance of the cells to peripheral vascular networks, and inherently decreases the oxygen tension delivered to the cells. In contrast, nanothin layer-by-layer (LbL) microencapsulation assembly has been proposed as a potential alternative technique for islet surface modification and which reduces the microcapsule thickness to approximately 35 nm [12]. This type of coating is advantageous as it employs alternate layers of polycations and polyanions, resulting in precise control over the thickness of the coating [13]. It has been proposed to use hydrogen bonds to assemble the LbL coating rather than electrostatic interactions due to associated cytotoxicity issues [12,14]. Recently, such an approach using poly(N-vinylpyrrolidone) (PVPON) and the antioxidant tannic acid (TA) to coat islets was demonstrated to facilitate euglycemia in diabetic mice post-transplant [15,16], delayed islet allo- and auto-immune rejection in mice [17] and reduce innate inflammatory immune signaling of encapsulated neonatal porcine islets [18]. The potential to apply such innovative approaches to modulate the immune responses evoked by implanted beta cells for clinical use is immensely promising. Therefore, we sought to examine the clinical relevance by testing the ability of nanothin PVPON/TA-coating to preserve human islets in vitro and in vivo function while conferring delayed rejection of MHC mismatch murine islet allografts.

Herein, our data demonstrate that PVPON/TA-coated murine and, for the first time, human islets retain their in vitro and in vivo functional potency, reduce inflammatory signaling, and delay allograft rejection. Taken together, LbL coating (PVPON/TA) may be an efficacious modality to protect current and future human beta cell replacement therapies.

## 2. Materials and Methods

### 2.1. Murine Islet Isolation

All studies were conducted in accordance with the institutional ethical committee of the University of Alberta (AUP0002977) and the Canadian Council of Animal Care. Pancreatic islets were isolated from 8- to 12-week-old male BALB/c mice (Jackson Laboratories) weighing between 20 and 25 g. Animals were housed under virus-antibody–free conditions having access to food and water ad libitum. Mice were maintained on a light/dark (12 h/12 h) cycle at 23 °C.

Mouse islets were isolated and purified based on our previously described methodology [19]. Briefly, the common bile duct was cannulated and the pancreata were distended with cold (4 °C) Liberase T-Flex (collagenase and thermolysin) enzyme (Roche Diagnostics, Laval, QC, Canada). Subsequently, distended pancreases were placed in a water bath (37 °C) with light shaking to free the islets from surrounding exocrine tissue. Islets were purified using Histopaque-density solutions (1.108, 1.083, and 1.069 g/mL, Sigma-Aldrich Canada Co., Ltd., Oakville, ON, Canada). Islets were washed with HBSS-supplemented fetal bovine serum (10%) and cultured in Connaught Medical Research Laboratories (CMRL-1066, Mediatech, Manasses, VA, USA) supplemented with fetal bovine serum (10%), L-glutamine (100 mg/L), penicillin (112 kU/L), streptomycin (112 mg/L), and HEPES (25 mmol/L) prior to encapsulation and transplantation.

### 2.2. Human Islets

Human pancreatic islets were isolated from deceased donors (*n* = 5) under ethical approval from the Human Research Ethics Board of the University of Alberta (Pro00092479) and obtained from the Clinical Islet Laboratory’s Distribution Program at Alberta Health Services (*n* = 2, male, 38–62 years old, BMI 24.3–26.9) and the Alberta Diabetes Institute’s Human Islet Core (*n* = 3, male, 42–57 years old, BMI 22.7–33.9). Deceased donor pancreata were processed for islet isolation, with informed consent obtained from the donor’s next of kin, and were not financially compensated.

### 2.3. Conformal Coating of Islets (PVPON/TA)

The coating, consisting of Poly(N-vinylpyrrolidone) (PVPON) (average MW 1,300,000 g/mol) (Fisher Scientific, Ottawa, ON, Canada), and tannic acid (TA), (MW 1700 g/mol) (Sigma-Aldrich Canada Co., Ltd., Oakville, ON, Canada), of pancreatic islets was performed in a biological safety cabinet at a room temperature in a dark condition. PVPON was the first layer adsorbed to the surface of the islet by continuous rotation of 5 mL tubes with islets in 4 mL PVPON solution 1 mg/mL (CMRL, pH = 7.4) for 8 min, followed by the deposition of TA layer 4 mL of 0.3 mg/mL (HBSS, pH = 7.4) for 8 min. Deposition of the next layers continued for 3 min. After each deposited layer, islets were collected by centrifugation for 1 min at 1000 rpm and washed with HBSS solution. Alternating polymer deposition onto islets continued until 3.5 layers were formed (4 layers of PVPON and 3 layers of TA) [12]. All solutions were filter-sterilized with polystyrene non-pyrogenic membrane systems (0.22 μm pore size) (Sigma-Aldrich Canada Co., Ltd., Oakville, ON, Canada) before use. PVPON/TA coating and morphological characterization of mouse and human islets were confirmed using transmission electron microscopy (TEM) and scanning electron microscopy (SEM) (FE-SEM, Hitachi, Tokyo, Japan).

### 2.4. In Vitro Islet Function Assessments

The functional capacity of the coated and uncoated islets was assessed through static in vitro glucose-stimulated insulin section (GSIS). Control and coated islets were incubated in CMRL solution before static incubation assay. Coated and uncoated islet samples were transferred to 6 mL low glucose (2.5 mM) for 1 h, followed by incubation in 6 mL high glucose (24.4 mM D-glucose) for 1 h. At the end of each incubation, conditioned media was collected, and insulin was measured by the ELISA for human insulin (ALPCO, Salem, NH, USA) and electrochemiluminescence for mouse insulin (MesoScale Discovery, Gaithersburg, MD, USA). Stimulation indices were calculated by dividing the amount of insulin released at 24.4 mM glucose by that released at 2.5 mM glucose. The vitality of the encapsulated and control islets was evaluated and quantified using the live/dead cell imaging fluorescence kit (Invitrogen, Thermo-Fisher, Ottawa, ON, Canada) previously described [20].

Oxygen consumption rates (OCRs) were examined for PVPON/TA and control islets from each islet isolation, using a fiber optic sensor (Instech Laboratories, Plymouth Meeting, PA, USA), as previously described [21]. OCR measurements were normalized to the quantity of DNA in each sample (nmol/min-mg DNA), by utilizing a Quant-iT PicoGreen dsDNA kit (Molecular Probes, Eugene, OR, USA).

### 2.5. Diabetic Induction

Approximately one week before transplant, recipient male and female C57BL/6 mice (approximately 20 weeks of age) (Jackson Laboratory) were rendered diabetic through the administration of an intraperitoneal injection of streptozotocin (STZ, Sigma-Aldrich Canada Co., Ltd., Oakville, ON, Canada) at 185 mg/kg freshly reconstituted in acetate phosphate buffer pH 4.5 [22]. Animals were considered diabetic and utilized for subsequent transplant studies after two consecutive blood glucose readings >18 mmol/L, measured using a OneTouch UltraMini glucose meter (LifeScan, Burnaby, BC, Canada).

### 2.6. Islet Transplantation

Human islets from each isolation were equally divided into two groups, coated and non-coated. For each isolation, 1500 human islet equivalents (IE) were implanted under the kidney capsule of diabetic male B6.129S7-Rag^1tm1Mom^/J mice (Rag^-/-^) (approximately 15 weeks) with either control (non-coated) (*n* = 4) or PVPON/TA-coated islets (*n* = 4). For allogeneic transplants, 500 islets  ±  10% BALB/c (H2^d^) islets were transplanted under the left renal capsule of diabetic C57BL/6 (H2^b^) mice [23]. Recipients were divided into four groups: (i) control (islets alone) (*n* = 9), (ii) PVPON/TA-coated islets (*n* = 10), (iii) CTLA4-Ig + islets (*n* = 12) and (iv) PVPON/TA + CTLA4-Ig + islets (*n* = 11). CTLA4-Ig (10 mg/kg, Biocell, West Lebanon, NH, USA) was injected i.p. on days 0, 2, 4, and 6 post-transplant. Cytotoxic T-lymphocyte–associated antigen 4 (CTLA-4)-Ig is a fusion protein that binds CD80/86 on antigen-presenting cells (APCs), preventing co-stimulatory interactions required for T cell activation [24], and is commonly used as a systemic immunosuppressive agent in our allograft models [19]. Nine separate islet isolations were conducted.

### 2.7. Evaluation of Islet Graft Function

After transplantation islet graft function was assessed through non-fasting blood glucose measurements, using a portable glucometer three times per week, in all groups transplanted. Diabetes reversal was defined as two consecutive readings <11.1 mmol/L. Allograft rejection was defined as two consecutive readings ≥18.0 mmol/L. In addition, to assess graft metabolic capacity in recipients that maintained normoglycemia at 30 days for human islets studies and at 60 days for allograft study post-transplant an intraperitoneal glucose tolerance test (IPGTT) was conducted. For the allograft study, naïve, nondiabetic C57BL/6 mice served as controls. After overnight fasting (~16 h), mice were injected intraperitoneally with 3 g/kg of glucose (DMVet, Coaticook, QC, Canada). Blood glucose measurements were collected at 0, 15, 30, 60, 90, and 120 min post-injection. Blood glucose area under the curve (AUC) values were analyzed and compared between transplant groups [25]. At 0 and 60 min time points, samples were taken for measuring human C-peptide levels by the ELISA (ALPCO, Salem, NH, USA). The stimulation index was calculated by dividing the amount of human C-peptide released at 60 min by that released at 0 min. Confirmation of graft-dependent euglycemia was conducted by survival nephrectomies and kidneys were fixed in 10% formalin (Thermo-Fisher, Ottawa, ON, Canada) for histological assessment.

### 2.8. Intra-Islet Graft Proinflammatory Cytokines

Separate acute allotransplant studies were conducted in which diabetic C57BL/6 mice were transplanted as described above and were monitored for 7 days. Animals were divided into three treatment groups: PVPON/TA-coated islets (*n* = 5), CTLA4-Ig + islets (*n* = 3), and PVPON/TA + CTLA4-Ig + islets (*n* = 3). CTLA4-Ig was administrated using the same protocol as described previously. Diabetic C57BL/6 mice transplanted with islets alone (*n* = 3) and naïve shame transplant mice (*n* = 3) were used as controls. Phosphate buffered saline (PBS) was injected for both control groups to mimic the experimental groups. Seven days post-transplant animals were nephrectomized and kidneys were lysed via homogenization and processed as previously described [19]. Subsequently, samples were analyzed with Mouse Pro-inflammatory 7-Plex Ultra-Sensitive Kit according to manufacture specifications (MesoScale Discovery) to measure intragraft proinflammatory cytokine profiles (IL-1β, IL-6, IFN-γ, TNF-α, IL-10, IL-12p70, KC/GRO).

### 2.9. Immunohistochemistry

Nephrectomized islet grafts were immediately fixed (Z-Fix VENDOR) for histological analysis. Islet grafts were stained for insulin and glucagon using our immunofluorescence methodology described previously [25]. Briefly, after rehydration, the graft sections were washed with PBS and blocked with 20% goat serum in PBS for 60 min. Subsequently, sections were stained with pig anti-insulin (Agilent, Santa Clara, CA, USA) and mouse anti-glucagon (Sigma-Aldrich) for 1 h. Following washing with PBS, sections were stained with secondary antibodies consisting of goat anti-guinea pig (AlexaFluor 488) and goat anti-mouse (AlexaFluor 594) for 1 hr at room temperature under dark conditions. Samples were then washed in PBS, followed by nuclei counter-staining with DAPI in an anti-fade mounting medium (ProLongGold, Thermofisher).

### 2.10. Statistical Analysis

Results are represented as mean ± standard error mean (SEM). In vitro data were analyzed using a two-tailed unpaired *t*-test. Allograft results were expressed as Kaplan–Meyer survival function curves and comparisons between groups analyzed by Log-rank (Mantel–Cox) test and two-ANOVA with multiple comparisons for the area under the curve for IPGTT and proinflammatory measurements. *p* < 0.05 was considered significant. All data were analyzed using GraphPad Prism version 9.0 statistical software (GraphPad Software, La Jolla, CA, USA).

## 3. Results

### 3.1. PVPON/TA-Encapsulated Murine Islets Maintain In Vitro Functionality

Transmission electron microscopy (TEM) and scanning electron microscopy (SEM) confirmed the conformal coating of human islets (Figure 1 and Figure 2). PVPON/TA coating did not increase the total transplant volume in comparison with non-coated islets (Control: 3.01 um^2^ ± 0.09 vs. PVPON/TA coated: 2.96 um^2^ ± 0.09, *p* > 0.05) (Figure 1A,D).

Islet viability was evaluated to determine if nanothin PVPON/TA coating influences islet function. Control and PVPON/TA-coated islets were handled under the same condition during coating, for instance, islets from both groups were rotated, washed, and centrifuged simultaneously. There was no significant difference between PVPON/TA-coated and non-coated mouse and human islets with respect to insulin release in response to glucose (GSIS) (Figure 3A,B), mitochondrial potency via oxygen consumption rate (OCR) (Figure 3C), and vitality via membrane integrity staining (Figure 3D–F), 24 h post-coating, indicating that a nanothin coating does not hinder the intrinsic metabolic in vitro function of both murine and human islets.

### 3.2. PVPON/TA-Encapsulation Human Islets Restore Glucose Control Post-Transplant

To investigate the impact of nanothin coating on in vivo function, 1500 human IE of PVPON/TA-coated and non-coated islets were transplanted under the kidney capsule of STZ-induced diabetic immunodeficient Rag^-/-^ mice. All transplanted mice became normoglycemic within 10 days and maintain normoglycemia until survival nephrectomies were performed 60 days post-transplantation. Upon removal of the islet graft, all recipients rapidly became hyperglycemic, confirming graft-dependent euglycemia (Figure 4A). IPGTT performed 30 days post-transplantation demonstrated a comparable glycemic response to a metabolic challenge (Figure 4B). No differences in blood glucose AUC values were observed between transplant groups (Figure 4C). Serum samples at 0 min (basal) and 60 min post-glucose administration (stimulated) were analyzed for human C-peptide. Stimulated C-peptide levels were significantly higher than basal levels in control recipients (*p* < 0.05, two-tailed *t*-test), but there was no significant difference between basal and stimulated C-peptide levels in the PVPON/TA group, which demonstrated higher basal C-peptide values (Figure 4D). The stimulated C-peptide index between control and PVPON/TA recipients did not differ (*p* > 0.05) (Figure 4E). Immunofluorescence staining of islet graft-bearing kidneys from both cohorts demonstrated similar islet graft morphology and stained positive for insulin (beta)- and glucagon (alpha)-secreting cells (Figure 4F). These data demonstrate that PVPON/TA-coated human islets maintain their ability to engraft and restore glucose control post-transplant.

### 3.3. PVPON/TA -Encapsulation Delays Allograft Rejection but Does Not Potentiate Systemic CTLA4-Ig Therapy

It has been demonstrated that the PVPON/TA coating preferentially modulates innate proinflammatory responses and delays auto and alloimmune responses in various murine transplant models [17,18,26]. We next wanted to assess the protective properties of PVPON/TA coating, as monotherapy and adjuvant to systemic co-stimulatory immunosuppression (CTLA4-Ig), in our MHC mismatch islet allograft model [19]. We transplanted 500 BALB/c islets under the kidney capsule of diabetic C57BL/6 mice, divided into four groups: non-coated islets alone (negative control), PVPON/TA-coated islets, islets + CTLA4-Ig, and PVPON/TA-coated islets + CTLA4-Ig. Islet allograft survival rates for recipients of PVPON/TA-coated islets (*n* = 10) were significantly higher than that of non-coated islet recipients (*n* = 9) (*p* < 0.01, log-rank) (Figure 5A), indicating that PVPON/TA coating delays alloimmune mediated rejection. As anticipated, PVPON/TA coating alone improved islet allograft survival; however, as monotherapy, it was insufficient to maintain long-term immunoprotection. By contrast, 4 of 12 (33.3%) CTLA4-Ig-alone recipients and 4 of 11 (36.4%) PVPON/TA-coated islets + CTLA4-Ig recipients demonstrated prolonged allograft function until recovery nephrectomy (Figure 5B). The similar rates of maintained euglycemia in these cohorts indicates that PVPON/TA coating did not potentiate the immunoprotective capacity of systemic CTLA4-Ig co-stimulatory blockade. To evaluate the glycemic response of transplanted islets to a metabolic challenge, euglycemic animals and naïve non-diabetic controls underwent an IPGTT at 60 days post-transplant (Figure 5C). All three groups showed similar glucose clearance rates (Figure 5D). These data suggest that PVPON/TA coating as monotherapy as well as an adjuvant to systemic immunosuppression delays alloimmune mediated islet rejection.

### 3.4. PVPON/TA-Encapsulation Modulates Intra-Islet Graft Inflammation

To characterize the localized immune phenotype evoked by PVPON/TA microencapsulation, acute transplants were conducted. Separate cohorts of transplant recipients were analyzed for localized proinflammatory cytokines (Figure 6; IL-1β, IL-6, IFN-γ, TNF-α) release 7 days post-transplant. The quantity of intra-islet graft IL-1β was reduced in both the CTLA4-Ig therapeutic groups, however, these data were not statistically significant (Figure 6A). IL-6 content was significantly lower in both CTLA4-Ig recipient groups compared to PVPON/TA-coated islet grafts (*p* < 0.05, Figure 6B). IFN-γ expression was significantly lower in CTLA4-Ig-alone recipients in comparison with islets-alone recipients (*p* < 0.05, Figure 6C). Similar to IL-6, intra-islet graft TNF-α concentrations were lower in islets + CTLA4-Ig and PVPON/TA-coated islets + CTLA4-Ig cohorts but not to a significant degree (Figure 6D). These data indicated that the prolonged allograft survival observed in the islets + CTLA4-Ig and PVPON/TA-coated islets + CTLA4-Ig recipients (Figure 5B), may be due to the reduction in acute proinflammation as measured 7 days post-transplantation.

## 4. Discussion

The etiology of type 1 diabetes (T1D) is associated with the immune-mediated destruction of pancreatic β cells. An effective therapeutic strategy to restore and maintain physiologic glycemic control, especially for patients with T1D who suffer from life-threatening hypoglycemia unawareness, is pancreatic islet transplantation. Beta cell replacement has become progressively more efficacious over the past two decades [8]. However, several limiting factors prevent islet transplantation from replacing exogenous insulin therapy, including the shortage of viable human organ donors and the need for chronic immunosuppression. Encapsulation strategies that prevent the immune destruction of transplanted beta cells, without systemic anti-rejection drugs, have the potential to increase patient inclusion. Recent macroencapsulation (e.g., Viacyte’s PEC-Encaptra) devices, housing stem-cell-derived-islet islets [27] or alginate islet microencapsulation approaches [28], are conceptually promising based on findings in rodents. However, the size of alginate-encapsulated islets can vary and exceed 800μm in diameter, increasing passive diffusion distances, and transplant volumes, and presenting a physical barrier impeding glucose-insulin metabolism and oxygen transportation leading to higher frequencies of cell death [29]. Furthermore, low-purity alginate and biomaterials can stimulate foreign body responses resulting in fibrotic overgrowth [30]. Recently developed purification technologies have been used that reduce alginate foreign body response [31,32,33]. Initial, clinical trials of stem-cell-derived islets housed in immunoprotective macroencapsulated devices (NCT02239354), conducted in a limited cohort of patients living with T1D, were halted due to fibrotic overgrowth. Recently, a series of follow-up clinical trials testing the same cell sources now housed in non-immunoprotective devices, engineered with porous membranes to allow direct vasculature exposure, improved beta cell survival and glucose-responsive insulin secretion, however, at the expense of requisite systemic immunosuppression to thwart auto- and allo-immune rejection [34,35]. These advancements indeed improve outcomes of beta cell replacement therapies, but the development and translation of immunosuppression-free transplant modality remain elusive.

An alternative to macro- and micro-encapsulation is layer-by-layer (LbL) nanothin coating of islets in biocompatible polymers. A promising LbL encapsulation methodology implements an immune inert polymer, poly(N-vinylpyrrolidone) (PVPON), and potent antioxidant tannic acid (TA) [16,17,36]. PVPON/TA layering can significantly reduce ROS and proinflammatory immune responses, and decrease T cell activation in vitro [15,16]. When tested in vivo, PVPON/TA encapsulation of mouse islets can skew macrophages toward an anti-inflammatory phenotype and delay immune rejection in NOD mice [17,18]. It has been suggested that a decrease in STAT1 signaling may be causal in shifting to an anti-inflammatory M2 macrophages phenotype, observed in PVPON/TA-encapsulated islet allografts [17]; however, the specific immunoprotective mechanism(s) elicited by PVPON/TA coating remains undefined. The immunomodulatory benefits of PVPON/TA coating are not restricted to murine islets, as Barra et al. demonstrated that PVPON/TA-coated neonatal porcine islets (NPI) suppressed xenogeneic proinflammatory innate immune responses, but, as anticipated, were unable to prevent xenograft rejection in NOD mice [18]. This innovative approach to modulating the immune responses evoked by implanted beta cells for clinical use is immensely promising. Therefore, for the first time, we sought to examine the clinical relevance of PVPON/TA coating to preserve human islets in vitro and in vivo function while conferring a delay in the rejection of MHC mismatch murine islet allografts.

We demonstrated that PVPON/TA-coated murine and human islets retain their in vitro and in vivo functional potency. Both human and murine PVPON/TA-coated islets demonstrated similar responses to in vitro glucose-stimulated insulin secretion compared to non-coated islets, in contrast to observations made with islets encapsulated in certain alginate-based modalities [11,37]; however, PVPON/TA-coated human islets did secrete more basal insulin compared to control islets. We further demonstrated that PVPON/TA does not compromise human and murine islet function, similar to the observation made with NPI [18], as evident through comparable OCR and membrane integrity measurements to that of non-coated islets. Our study is the first to demonstrate PVPON/TA-coated human islets maintain their in vivo function, as immunocompromised recipients of PVPON/TA-coated human islets reverse diabetes and maintain euglycemia until islet graft removal similarly to non-coated islet recipients. Glucose clearing and human C-peptide levels during an IPGTT also were similar between coated and non-coated groups, supporting the notion that PVPON/TA coating does not hinder human islet engraftment and transplant function. Our data also supports expanding the utility of PVPON/TA coating to clinically approved human stem cell-derived islet-like clusters [34,35,38] in future studies. However, it has been well-documented that the aggregation of human islet amyloid polypeptide (hIAPP), a major regulatory peptide secreted with insulin by the beta cell [39], is associated with causal beta cell loss in type 1 and 2 diabetes and in pancreatic islets post-transplantation [40]. Even though we demonstrated that PVPON/TA-encapsulated human islets retain their ability to secrete insulin in a glucose-dependent manner, we did not examine if IAPP aggregates within coated human islets. Therefore, future studies should be directed toward examining the functional impact of hIAPP in PVPON/TA-coated human islets, both in vitro and in vivo.

Corroborating the findings by Barra et al. [17], we too observed that PVPON/TA coating as monotherapy can delay murine MHC-mismatch allograft rejection compared to recipients of non-coated islets, however, long-term euglycemia was not durable. Herein, encapsulated islets demonstrated no defects in glucose responsiveness and similar transplant efficacy results in comparison with other experimental groups. Therefore, we speculated that combining PVPON/TA coating as an adjuvant with other immunomodulatory modalities would potentiate allograft protection. Cytotoxic T lymphocyte-associated antigen 4-Ig (CTLA4-Ig) subverts T-cell activation by blocking costimulatory signaling pathways [41] and, similar to our observations, alone yields prolonged murine islet allograft survival [23]. Acute transplants demonstrated a decrease in IL-6, despite no changes in gene expression, and IL-1β levels in islets + CTLA4-Ig and PVPON/TA-coated islets + CTLA4-Ig in comparison with the PVPON/TA group. This effect might be mainly driven by CTLA4 Ig, as it was previously shown that CLTA4-Ig decreased IL-6 levels in rheumatoid synovial macrophages in the monoculture [42] and human B cells [41]. IFN-γ expression was significantly lower in islets + CTLA4 Ig in comparison with islets alone. However, the kinetics and influence of cytokine and chemokine expression within PVPON/TA grafts could differ at earlier time points post-transplant. Systemic CTLA4-Ig administration may impact T cells and accelerate rejection, depending on details of the injection timing and degree of mismatch [43,44,45]. In long-term follow-up studies, we observed that PVPON/TA-coated BALB/c islets transplanted into diabetic C57Bl/6 mice administered acute (D0, 2, 4, and 6) systemic CTLA4-Ig didn’t show a significant difference in euglycemia duration in comparison to systemic CTLA4-Ig alone. This might be explained, in part, by their activity towards similar rejection pathways.

Since PVPON/TA coatings can be modified to increase the number of layers of PVPON and TA for encapsulation, tethering with immune inhibitory drugs, including PD-L1, rapamycin, or tacrolimus, might further enhance localized immunosuppression. Furthermore, the pretreatment of islets with cytoprotective agents targeting apoptosis (e.g., caspase inhibitors) [46], endoplasmic reticulum (ER) stress (e.g., necrostatins) [47], and growth factors (e.g., fibroblast growth factors) [48] prior to encapsulation may be an effective strategy to improve long-term functional outcomes. Supporting this notion and in contrast to our observation with systemic CTLA4-Ig, Barra et al. recently demonstrated that, when recombinant CTLA-4-Ig was conjugated to the outer surface of PVPON/TA layers, PVPON/TA/CTLA-4-Ig-encapsulated islet allografts maintained function significantly longer than non- and (PVPON/TA/IgG)-encapsulated islets. The addition of CTLA-4-Ig induced an expansion of CD25+ FoxP3+ Tregs, anergic CD4+ T cells, and enhanced IL-10 production [26]. Future studies will investigate how graft-localized drug-eluting micelles [19] may also enhance the localized immunosuppression elicited by PVPON/TA. Additionally, in contrast to macro- and micro-encapsulated approaches, PVPON/TA coating did not significantly increase the volume of human islet transplant, thereby allowing for potential intraportal infusion or transplantation into extrahepatic sites such as prevascularized subcutaneous tissue [25], bioabsorbable scaffolds [49], or omentum [50].

## 5. Conclusions

Collectively, the results of this study demonstrate that PVPON/TA does not increase the total transplant volume of human islets and does not hinder human islets’ in vitro and in vivo function. PVPON/TA delays murine allograft rejection in MHC mismatching islet allografts; however, it does not potentiate the therapeutic benefit of systemic CTLA4-Ig therapy. Future studies will be focused on increasing murine islet allograft survival by combining PVPON/TA coating with localized immunosuppression modalities and clinically relevant transplant sites.

## Figures and Tables

**Figure 1 pharmaceutics-15-01137-f001:**
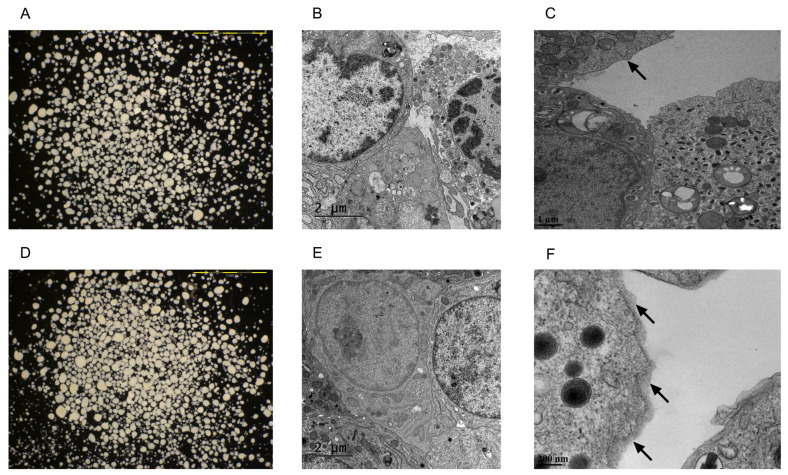
Photomicrographs and transmission electron microscopy images of non-coated (**A**–**C**) and PVPON/TA-coated (**D**–**F**) human islets. The scale bars are 1 mm in both light microscopy images. Non-coated (**B**) and PVPON/TA-coated (**E**) human islets were transplanted under the kidney capsule of B6.129S7-Rag^1tm1Mom^/J mice. Non-coated (**D**) and PVPON/TA-coated (**F**) human islets one day post-coating. The arrows point to the edges of the islet.

**Figure 2 pharmaceutics-15-01137-f002:**
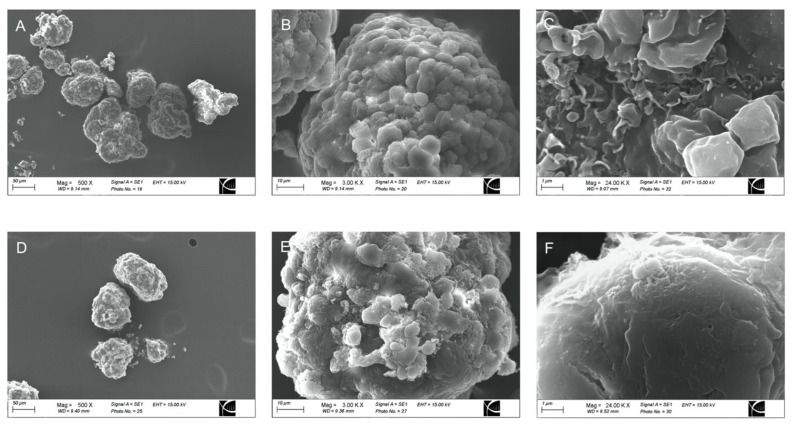
Representative scanning electron microscopy micrograph of non-coated (**A**–**C**) and PVPON/TA-coated (**D**–**F**) human islets at 500× (**A**,**D**), 3000× (**B**,**E**), and 24,000× magnification (**C**,**F**).

**Figure 3 pharmaceutics-15-01137-f003:**
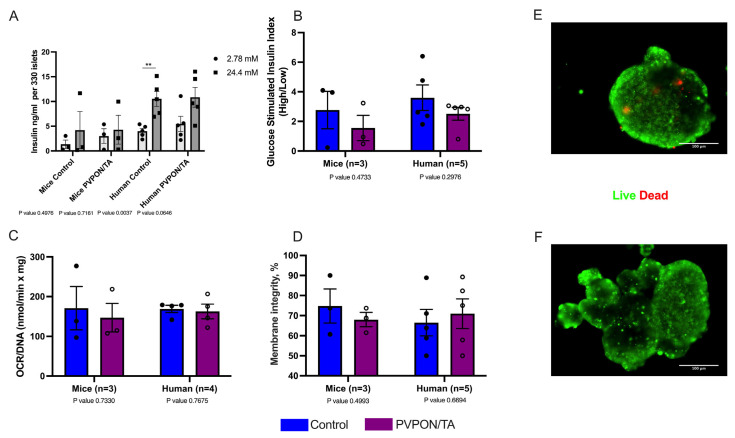
In vitro function evaluations of non-coated (control-blue) and PVPON/TA-coated (purple) mouse and human islets. Glucose stimulated insulin levels (GSIS) (** *p* < 0.01, two-tailed *t*-test) (**A**) and stimulation index (**B**) from non-coated and coated mouse (*n* = 3 isolations) and human islets (*n* = 5 human donors). Oxygen consumption rates (OCR/DNA) of non-coated and PVPON/TA-coated mouse and human islets (**C**). Comparison of vitality measured by membrane integrity of non-coated and PVPON/TA-coated mouse and human islets (**D**). Representative live (green) and dead (red) membrane staining of non-coated (**E**) and PVPON/TA-coated (**F**) human islets. Data represented as mean ± SEM. Scale bars represent 100 μm.

**Figure 4 pharmaceutics-15-01137-f004:**
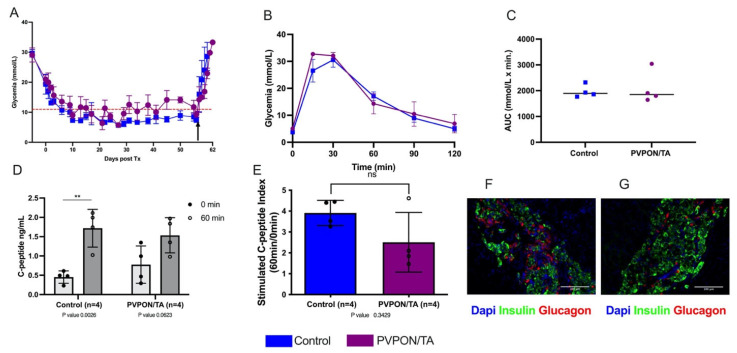
PVPON/TA-coated human islets retain their ability to restore euglycemia post-transplant. Post-transplant non-fasting blood glucose measurements of Rag^-/-^ mice transplanted with either 1500 non-coated (*n* = 4) or PVPON/TA-coated (*n* = 4) human islets equivalent (**A**). Arrow represents survival nephrectomy. Blood glucose profiles (**B**) and AUC (**C**) following intraperitoneal glucose tolerance testing 30 days post-transplant. Animals from the control group (*n* = 4, blue) demonstrated similar glucose clearance compared to PVPON/TA-coated islet recipients (*n* = 4, purple) (*p* > 0.05, two-tailed *t*-test). During the glucose tolerance test, samples were taken for human C-peptide levels (**D**). C-peptide basal and stimulated measurements from the control group were significantly different (** *p* < 0.01, two-tailed *t*-test). C-peptide levels from PVPON/TA recipients did not differ between basal and stimulated levels. Stimulated C-peptide index similar between control and PVPON/TA recipients (**E**). Immunohistochemistry of human islet PVPON/TA-coated (**F**) and control (**G**) grafts. Data represented as mean ± SEM. Scale bars represent 100 μm.

**Figure 5 pharmaceutics-15-01137-f005:**
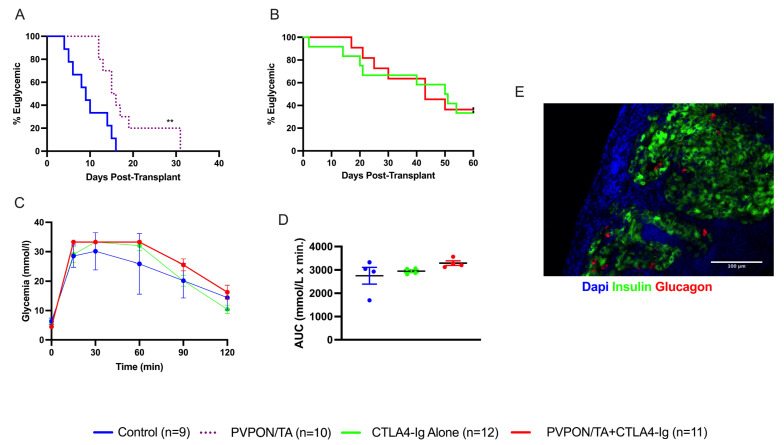
Transplantation of PVPON/TA-coated murine islets delays allograft rejection. Kaplan–Meier Log-rank survival curve of STZ-treated C57BL/6 recipients transplanted with 500 BALB/c control islets (blue, *n* = 9) or PVPON/TA-coated islets (purple dash, *n* = 10) under the kidney capsules (**A**). Kaplan–Meier Log-rank survival curve of STZ-treated C57BL/6 recipients transplanted with 500 BALB/c islets + CTLA4-Ig (green, *n* = 12) or PVPON/TA-coated islets + CTLA4-Ig (red, *n* = 11) under the kidney capsule (**B**). Intraperitoneal glucose tolerance test (IPGTT) on allogeneic euglycemic recipients 60 days post-transplant for naïve non-diabetic mice (blue, *n* = 4), islets + CTLA4-Ig (green, *n* = 4), PVPON/TA-coated islets + CTLA4-Ig (red, *n* = 4) (**C**). Two-way ANOVA with multiple comparisons for the area under the curve (AUC) of the 60-day IPGTT (**D**). Immunohistochemistry of mouse allograft transplanted with PVPON/TA-coated islets + CTLA4-Ig, explanted 60 days post-transplant (**E**). ** *p* < 0.01.

**Figure 6 pharmaceutics-15-01137-f006:**
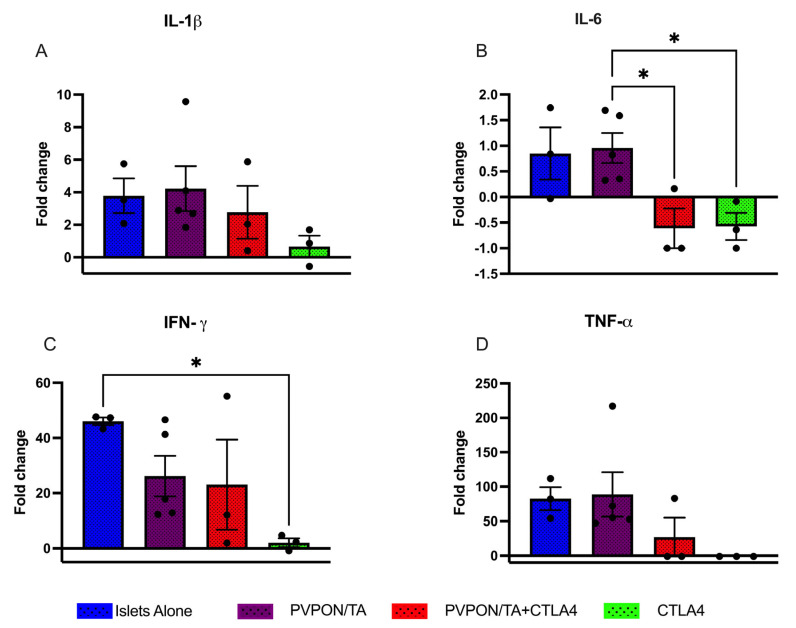
Intra-islet graft proinflammatory cytokines analysis of IL-β (**A**), IL-6 (**B**), IFN-γ (**C**), TNF-α (**D**) from murine allografts recipients procured 7 days post-transplant. C57BL/6 mice were transplanted with BALB/c islets (~500 islets) under the kidney capsule. Mouse recipient groups included islets alone (*n* = 3, blue), PVPON/TA-coated islets (*n* = 5, purple), PVPON/TA-coated islets+ CTLA-4-Ig (*n* = 3, red), and Islets + CTLA4 Ig (*n* = 3, green). Nondiabetic, non-transplanted C57BL/6 mice (*n* = 4) served as control. Data expressed as a fold change from control grafts. Data points are represented as mean ± SEM. * *p* < 0.05.

## Data Availability

All datasets and analyses contained within this study can be obtained from the corresponding author upon reasonable request.

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
