# Peer review of "Nanothin Conformal Coating with Poly(N-vinylpyrrolidone) and Tannic Acid (PVPON/TA) Preserves Murine and Human Pancreatic Islets Function"

_pharmaceutics, 2023, doi:10.3390/pharmaceutics15041137_

Round 1

Reviewer 1 Report

The paper by Polishevska et al examined the conformal coating (poly(N-vinylpyrrolidone and tannic acid) of islet cell for transplantation with respect to being functional and reducing inflammation/ immune reaction.  The study conducts in vitro evaluation of coated and non-coated islets, as well as two in vivo phases - human coated and non-coated islets into diabetic immunodeficient mice and BALB/c islets from mice into diabetic C57BL/6 mice.  Islet function was determined via a variety of measures, as well as inflammation (i.e., determination of pro-inflammatory cytokines).  Collectively, the manuscript provides solid evidence that coated islets were functional both in vitro and in vivo.  They also demonstrate that coated cells can delay allograft rejection compared to non-coated cells.  This manuscript has tremendous translational relevance to advance the field of pancreatic islet cell transplantation for type 1 diabetic patients.

Author Response

Many thanks for your and the review committee’s most thoughtful and constructive comments to improve our manuscript, "Nanothin Conformal Coating with Poly(N-vinylpyrrolidone) and Tannic Acid (PVPON/TA) Preserves Murine and Human Pancreatic Islets Function" (pharmaceutics-2270130). We have extensively revised the manuscript to incorporate all of the suggestions (see ‘Track-Changes’), and we very much appreciate the thoughtful input of the review team.

Reviewer 1 comments: The paper by Polishevska et al examined the conformal coating (poly(N-vinylpyrrolidone and tannic acid) of islet cell for transplantation with respect to being functional and reducing inflammation/ immune reaction.  The study conducts in vitro evaluation of coated and non-coated islets, as well as two in vivo phases - human coated and non-coated islets into diabetic immunodeficient mice and BALB/c islets from mice into diabetic C57BL/6 mice.  Islet function was determined via a variety of measures, as well as inflammation (i.e., determination of pro-inflammatory cytokines).  Collectively, the manuscript provides solid evidence that coated islets were functional both in vitro and in vivo.  They also demonstrate that coated cells can delay allograft rejection compared to non-coated cells.  This manuscript has tremendous translational relevance to advance the field of pancreatic islet cell transplantation for type 1 diabetic patients.

Response: We appreciate your valuable time and effort in reviewing our manuscript. Thank you for your kind feedback and enthusiasm for our work. We incorporated changes to reflect the suggestions provided. We have made moderate English, grammatical and spelling changes throughout the revised manuscript, as advised.

Reviewer 2 Report

In “Nanothin Conformal Coating with Poly(N-vinylpyrrolidone) and Tannic Acid (PVPON/TA) Preserves Murine and Human Pancreatic Islets Function, Polishevska et al, evaluated if the conformal coating of islets, with 22 poly(N-vinylpyrrolidone) (PVPON) and tannic acid (TA) (PVPON/TA), could preserve murine and human islet function while conferring islet allograft protection. Authors found that PVPON/TA-coated and control human islets were able to restore euglycemia post-transplant. Although PVPON/TA-encapsulation delays allograft rejection but does not potentiate systemic CTLA4-Ig therapy. Likewise, there are several points:

  1. In figure 5, about the glycemic response of transplanted islets underwent an IPGTT at 60-days post-transplant, it is not possible to visualize the potential effects of treatment with PVPON/TA-coating. The authors would complete the discussion or show evidence of this result.
  2. Authors evaluated the secretion of Peptide-C, however, is broadly described hIAPP is co-secreted with insulin, and moreover, this characterization could be desirable.
  3. Authors evaluated peptide-C secretion; however, it is broadly described that the hormone IAPP is co-secreted with insulin. Indeed. IAPP is correlated with Beta-cells functionality. This characterization could be desirable.
  4. Considering the high protein-secretory activity in islets, cell phenomena such as endoplasmic reticulum (ER) stress could be dysregulated, even more, triggering apoptosis. It is possible that ER stress could be a critical phenomenon.
  5. Although PVPON/TA-encapsulation delays allograft rejection, authors could consider a pre-treatment of islets (v. gr, growth factors) to improve the beneficial metabolic effect in the long term.
  6. In the manuscript is desirable to improve the resolution of the figures.

Author Response

Many thanks for your and the review committee’s most thoughtful and constructive comments to improve our manuscript, "Nanothin Conformal Coating with Poly(N-vinylpyrrolidone) and Tannic Acid (PVPON/TA) Preserves Murine and Human Pancreatic Islets Function" (pharmaceutics-2270130). We have extensively revised the manuscript to incorporate all of the suggestions (see ‘Track-Changes’), and we very much appreciate the thoughtful input of the review team.

Reviewer 2 comments: In “Nanothin Conformal Coating with Poly(N-vinylpyrrolidone) and Tannic Acid (PVPON/TA) Preserves Murine and Human Pancreatic Islets Function, Polishevska et al, evaluated if the conformal coating of islets, with 22 poly(N-vinylpyrrolidone) (PVPON) and tannic acid (TA) (PVPON/TA), could preserve murine and human islet function while conferring islet allograft protection. Authors found that PVPON/TA-coated and control human islets were able to restore euglycemia post-transplant. Although PVPON/TA-encapsulation delays allograft rejection but does not potentiate systemic CTLA4-Ig therapy. Likewise, there are several points:

Responses: Thank you for your constructive and valuable feedback. We very much appreciate the time and effort you have put into your comments on our manuscript. We have addressed each critique below:

1) In figure 5, about the glycemic response of transplanted islets underwent an IPGTT at 60-days post-transplant, it is not possible to visualize the potential effects of treatment with PVPON/TA-coating. The authors would complete the discussion or show evidence of this result.

Thank you for pointing this out. We agree with this comment. We emphasized it in a paper and comment about this in the revised Discussion lines 874-875. IPGTTs were conducted on 60th-day post-transplant on euglycemic animals, only. Animals from PVPON/TA and Control groups were rejected before 60th-day post-transplant, therefore it is not possible to visualize the potential effects of coating (as they were rejected). As an additional control for IPGTT, we used naïve non-diabetic mice. We highlighted it in the manuscript (Figure 5, legend). Please see page 12, line 762 “To evaluate the glycemic response of transplanted islets to a metabolic challenge, euglycemic animals and naïve non-diabetic controls  underwent an IPGTT at 60-days post-transplant (Figure 5C).” and page 13, Figure 5 legend “Intraperitoneal glucose tolerance test (IPGTT) on allogeneic euglycemic recipients 60 days post-transplant for naïve non-diabetic mice (blue, n=4), islets + CTLA4-Ig (green, n=4), PVPON/TA-coated islets + CTLA4-Ig (red, n=4) (C).”

2) Authors evaluated the secretion of Peptide-C, however, is broadly described hIAPP is co-secreted with insulin, and moreover, this characterization could be desirable.

Thank you for this intriguing comment. We did not observe any hindrance in insulin secretion post-human islet encapsulation with PVPON/TA, and would anticipate no alterations in hIAPP secretion. However, we did not examine if hIAPP aggregates post-encapsulate. Future studies will certainly be directed towards examining if hIAPP and its aggregation have a functional impact on PVPON/TA-coated human islets both in vitro and in vivo (e.g. thioflavin staining). We have revised this Discussion and added two new references to address this important point (Lines 855-871).

3) Authors evaluated peptide-C secretion; however, it is broadly described that the hormone IAPP is co-secreted with insulin. Indeed. IAPP is correlated with Beta-cells functionality. This characterization could be desirable.

See the answer to Question 2. 

4) Considering the high protein-secretory activity in islets, cell phenomena such as endoplasmic reticulum (ER) stress could be dysregulated, even more, triggering apoptosis. It is possible that ER stress could be a critical phenomenon.

Thank you for your perceptive comment and suggestion. We completely agree and have modified the Discussion (Lines 892-895) and included additional references to highlighting the potential cytoprotective benefits of this approach.

5) Although PVPON/TA-encapsulation delays allograft rejection, authors could consider a pre-treatment of islets (v. gr, growth factors) to improve the beneficial metabolic effect in the long term.

Thank you for your insightful comment. We agree and have modified the Discussion (Lines 884-886) and included 2 additional references highlighting the potential metabolic benefits of this approach.

6) In the manuscript is desirable to improve the resolution of the figures.

We have increased the resolution of the Figures to 1200 dpi and uploaded a revised Figure 3 to better illustrate islet live/dead staining.